

# A Bayesian Approach to Historical Climatology for the Burgundian Low Countries in the 15th Century

Chantal Camenisch[1,2], Fernando Jaume-Santero[1,3,4], Sam White[5], Qing Pei[6], Ralf Hand[1,7], Christian Rohr[1,2], and Stefan Brönnimann[1,7]

[1]Oeschger Centre for Climate Change Research, University of Bern, Switzerland
[2]Institute of History, Section of Economic, Social and Environmental History, University of Bern, Switzerland
[3]Department of Radiology and Medical Informatics, University of Geneva, Switzerland
[4]Haute école de gestion de Genève, University of Applied Sciences Western Switzerland, Switzerland
[5]Department of History, Ohio State University, Columbus, USA
[6]Department of Social Sciences, Education University of Hong Kong, China
[7]Institute of Geography, Climate Unit, University of Bern, Switzerland

**Correspondence:** Chantal Camenisch (chantal.camenisch@hist.unibe.ch)

**Abstract.**

Although collaborative efforts have been made to retrieve climate data from instrumental observations and paleoclimate records, there is still a large amount of valuable information in historical archives that has not been utilized for climate reconstruction. Due to the qualitative nature of these datasets, historical texts have been compiled and studied by historians aiming to describe the climate impact in socio-economical aspects of human societies, but the inclusion of this information in past climate
reconstructions remains fairly unexplored. Within this context, we present a novel approach to assimilate climate information contained in chronicles and annals from the 15th century to generate robust temperature and precipitation reconstructions of the Burgundian Low Countries, taking into account uncertainties associated with the descriptions of narrative sources. After data assimilation, our reconstructions present a high seasonal temperature correlation of ∼0.8 independently of the climate model employed to estimate the background state of the atmosphere. Our study aims to be a first step towards a more quantitative use
of available information contained in historical texts, showing how Bayesian inference can help the climate community with this endeavour.

## 1 Introduction

Historical texts, both descriptive sources, such as chronicles, and written records of phenology, such as grape harvest dates,
have enabled high-resolution reconstructions of temperature and precipitation for periods prior to the modern instrumental record (White et al., 2018). The principal approach for these reconstructions has been the "index" method (Pfister et al., 2018; Nash et al., 2021). Historical climatologists working with these texts have converted their information into ordinal indices (typically -3 to +3), which approximate departures from average monthly or seasonal conditions, and they have calibrated these indices to early instrumental data to obtain reconstructed values for the pre-instrumental period. The resulting temperature
and precipitation reconstructions have demonstrated high reconstruction skill, especially at the decadal scale. Nevertheless,



the index method has significant drawbacks, including loss of low-frequency variability and gaps in the reconstructions for seasons without descriptions, particularly in regions and periods with less comprehensive historical records such as those from the Middle Ages or Early Modern Times (Brázdil et al., 2010; Pfister et al., 2018).

Moreover, the index method does not capture the full range of inferences obtainable from the analysis of historical records. For example, if a researcher in historical climatology found reliable descriptions that a winter was unusually cold, she might assign that winter a value of -2. However, the "-2" would express neither the uncertainty regarding that value nor the possibilities of other values. While she is likely to encounter such descriptions for a cold winter (-2 on the index scale), she may also to find them for an extremely (-3) or slightly (-1) cold winter, but would very rarely find them for a mild or warm winter (+1
through +3). The index method does not express this range of likelihoods over different values. Similarly, the absence of any historical description is more likely for normal seasons than extreme seasons, but the index method cannot assign any value where there is a gap in the evidence.

To address these problems, this article presents a novel Bayesian approach for historical climatology. Bayesian approaches
have been widely adopted in other fields of climate science, including paleoclimate reconstructions, reanalysis, and integration of written descriptions with physical proxies (e.g., Brönnimann et al., 2013; Luterbacher et al., 2016; Salinas et al., 2016; Gennaretti et al., 2017; Osman et al., 2021). Here, we develop a Bayesian method for seasonal temperature reconstructions and apply it to a sample of data from the 15[th] century Burgundian Low Countries (Camenisch, 2015a, b). Our approach builds on existing familiarity and expertise with the index method. It uses climate models to set prior probabilities for each index value,
and instructs historical climatologists to assign likelihoods for the evidence (or lack thereof) given each index value. Applying Bayes' theorem, these likelihoods are then used to obtain updated posterior probability distributions.

Our Bayesian method has the following goals: (1) to distinguish and integrate the disciplinary expertise of paleoclimatologists, climate modelers, and historical climatologists; (2) to extend the useful application of historical climatology to regions
and periods with less abundant or consistent records; (3) to ensure that the maximum possible information can be applied to obtaining posterior probabilities; (4) to incorporate the absence of evidence as a probabilistic indicator of conditions; (5) to ensure more continuous reconstructions which include low-frequency variability; and (6) to improve reconstruction skill.

## 2   Data

In this study, we used two different groups of sources to reconstruct the climate of the Burgundian Low Countries. The first
source (Subsection 2.1) includes documentary data from historical archives describing past climate events of the study region that have been rescued, compiled, and studied by historians in previous publications (e.g., Camenisch, 2015b), while the second one (Subsection 2.2) is composed of temperature and precipitation simulations from large ensembles of General Circulation Models (GCM).



## 2.1 Documentary data

The basis for the reconstruction used here are primarily descriptions of weather from narrative sources - in the 15[th] century these were mainly chronicles and annals. These texts were written by clerics and laymen in Latin or a vernacular language. In the Burgundian Low Countries, chronicles and annals were thus written in Latin as well as in medieval forms of Flemish, Low German, Walloon, and other French dialects (Camenisch, 2015a). Many of the descriptions contained in these sources report directly on weather patterns, while others include comprehensive descriptions of societal climate impacts. These sources are 60 supplemented by municipal accounts, which also contain brief references to weather patterns, and by customs accounts, which Buisman (1996, 1998) evaluated for his climate reconstruction. The number of weather descriptions is not equally distributed within the century. In general, years with extreme weather events have significantly more reports than more normal years. However, this is not necessarily the case in less documented phases of the century. Such phases are the 1450s and 1460s, when generally fewer weather sensitive sources are available for the study area.


  One of the advantages of these sources, however, is that they cover the entire year, with almost similar numbers of descriptions falling on winter, spring, and summer, and somewhat fewer on autumn. The sources contain information on both temperatures and precipitation.

In addition to the actual weather descriptions, however, the metadata for each source and for each individual record also play a major role for the Bayesian approach. These metadata comprise the information that would be called extended "source criticism" in the historical sciences. This includes, for example, biographical information on the authors of the texts in order to evaluate whether they were eyewitnesses to the events described, or the number and reliability of the sources of a particular year or season.

## 2.2 Model ensembles

  The Bayesian approach requires a prior that is based on a prior knowledge. Climate model simulations can provide prior probability estimates based on physical constraints (see Subsection 3.3), information on external forcings (e.g., volcanic eruptions), or even effects of large-scale oceanic variability modes such as El Niño/Southern Oscillation. We used two sets of simulations to generate two priors: atmospheric simulations that were constrained, among other factors, with time-varying sea-surface tem-80 peratures from reconstructions as well as coupled simulations in which only the external forcings are prescribed (see Table 1). Figure 1 shows grid-point centres within the the Burgundian Low Countries where temperature and precipitation series were extracted.

  In first place we have Historical simulations for Palaeo-RA - Past 600 years with Observed forcings (HIPPO) which are 85 composed of 40 full-forcing members of the ECHAM 6.3 model with T63 horizontal and L47 vertical resolutions. Sea Surface Temperatures (SST) and Sea Ice Conditions (SIC) have been prescribed from either Samakinwa et al. (2021) or HadISST2





(Titchner and Rayner, 2014), using for the latter 10 different realizations so that SST and SIC uncertainties are taken into account. Vegetation and Land Use / Land Change (LULC) have also been prescribed. Moreover, almost all radiative forcings have been selected following CMIP6 / PMIP4 past2k specifications (Jungclaus et al., 2017) with standard PMIP4 volcanic

forcings (e.g., Sigl et al., 2015; Toohey and Sigl, 2017) for the first 20 members and Samakinwa et al. (2021) for the remaining members of the ensemble.

Simulations from the Community Earth System Model - Last Millennium Ensemble (CESM-LME) have also been used to test the independence of our reconstructions from the model employed. The CESM-LME Project (Otto-Bliesner, 2016)

includes 13 simulations of the CESM 1.1 CAM5 with a regular resolution of $1.9° \times 2.5°$, 30 vertical levels, and all transient forcings (i.e. solar radiation, volcanic aerosols, greenhouse gases, LULC, and orbital parameters). Both ensembles have their own advantages and disadvantages. For instance, HIPPO simulations begin in 1420 CE while the CESM-LME starts in 850 CE, allowing for the reconstruction of the entire 15[th] century. On the other hand, the HIPPO ensemble is composed of 40 members in contrast with the 13 members of the CESM-LME, which makes the former a more robust candidate to describe

the background state of the atmosphere.

Two climate variables at seasonal temporal resolution (i.e., DJF, MAM, JJA, SON) have been used from the HIPPO ensemble: air temperature at 2 meters (temp2) and wet days in a month (wetdaysinmonth). While temp2 is a standard output of the model, wetdaysinmonth is a custom variable defined as the number of days with more than 1 mm precipitation within a

model day. Note that we decided to use this customized variable as a precipitation estimator because it presents a Gaussian distribution, which it is suitable for our reconstruction method (see Subsection 3.4). Moreover, seasonal 2-meter air temperatures have also been extracted from the CESM-LME (TREFHT).

## 3   Methodology

Historical documents have been assimilated by using the famous Bayes' theorem (e.g., Efron, 2013; Puga et al., 2015) to update

prior atmospheric states estimated from model ensembles following Equation 1,

$$\mathbf{p}(Climate \,|\, Observation) = \mathbf{p}(Climate) \cdot \frac{\mathbf{p}(Observation \,|\, Climate)}{\mathbf{p}(Observation)} \tag{1}$$

where $\mathbf{p}(Climate \,|\, Observation)$ and $\mathbf{p}(Climate)$ represent the posterior and prior probability distributions, while $\mathbf{p}(Observation \,|\, Climate)$ is the probability distribution associated with the information from historical documents. Note that the right term is

divided by a normalizing constant (Eq. 2), which ensures that the sum of posterior probabilities add up to 1.



$$\mathbf{p}(Observation) = \sum_{Obs.=-3}^{3} \mathbf{p}(Climate) \cdot \mathbf{p}(Obs. \mid Climate) \tag{2}$$

Figure 2 illustrates the methodology followed to obtain the climate reconstruction of the Burgundian Low Countries. This novel approach uses the probability distribution of Pfister indices (see Subsection 3.1) to quantify the information from his-

torical archives (i.e. notes and qualitative observations preserved in documents and books) and, at the same time, assess the uncertainty associated with those same documents in terms of source reliability, data quality, and regional impact (see Subsection 3.2). In addition, a forward model has also been employed to convert ensemble members into Pfister indices, transferring the information contained in the background state of the atmosphere into prior probability distributions (see Subsection 3.3). Finally, the posterior probability is obtained using Bayes' theorem (see Subsection 3.4).

### 3.1 Pfister indices

Climate indices named after Christian Pfister serve as the basis for this reconstruction (e.g., Pfister et al., 1999, 2018). Temperature and precipitation are reconstructed separately. Depending on the source density, a seasonal, monthly, or other temporal resolution is chosen. Due to the characteristics of the sources and their density, a seasonal resolution was chosen for this present reconstruction. The individual indices have a 7-degree scale, which contains the following index values with regard to tem-

peratures (Fig. 3): -3=extremely cold, -2=very cold, -1=cold, 0=normal, +1=warm, +2=very warm, +3=extremely warm. For the precipitation indices, the scale is adjusted accordingly: -3=extremely dry, -2=very dry, -1=dry, 0=normal, +1=wet, +2=very wet, +3=extremely wet. In order to be able to determine the actual index value, a list of events and characteristics was developed for each individual index, which must apply to a season for it to be assigned to a specific index value. These properties are determined based on the climatic conditions known today in the study area and adapted as far as possible to the conditions of

the 15[th] century. For example, a winter -3 in the temperature index requires large watercourses to carry ice cover, or a summer can only be assigned an index value of -3 in the precipitation reconstruction if springs as well as smaller watercourses dried up and the grain harvest was negatively affected (e.g., Camenisch, 2015a, b). Ice formation on water bodies occurred more frequently before the large straightening and channelization projects or before the discharge of industrial wastewater, so the 15[th]-century report of (today rather rare) drift ice does not constitute an extremely cold winter (-3) in the 15[th]-century Low

Countries. In many cases, however, the assignment is not so clear that for one season the other 6 index values could be clearly excluded. Until now, it has not been possible to express this uncertainty or the probability with which the season is assigned to a particular index value. For this reason, we have taken the scale of the Pfister indices as a basis and have further developed this approach.

### 3.2 Inclusion of probabilities and uncertainties

Because in many seasons the descriptions in the sources do not allow a simple and unambiguous assignment to an index value, reconstructions with the index method have so far been forced to settle on the most probable index value. The discussion of





the reliability of a source, the assessment of uncertainties, and the weighing of the probability of a specific finding normally employed by historians and expressed in descriptive texts in the historical sciences were lost by this (allegedly) unambiguous decision in favour of an index value.


With the Bayesian approach presented here, it is possible to express this information in terms of a distribution of likelihoods. The method used here is again based on Pfister indices, but it now requires that the researcher assign a likelihood of getting such as observation assuming that the true climate matched each value in the scale. In total, these likelihoods must again add up to exactly 1. Moreover, each likelihood should be at least 0.05, since even the best source criticism cannot rule out misdating

of observations.

For example, let us say that the sources for a particular summer describe warm, sunny weather, a good harvest, and average or low prices for agricultural products. Using the conventional historical method, the historical climatologist would probably assign the summer a +2 on the Pfister scale, on the assumption that this is the most probable state of the climate given the

observations: $\mathbf{p}(Climate \,|\, Observation)$. The Bayesian approach, on the other hand, would consider the likelihood of the evidence assuming each value for the climate: $\mathbf{p}(Observation \,|\, Climate)$. In this case, the observations would be very unlikely assuming an extremely cold (-3) or very cold (-2) summer and also unlikely assuming a cold (-1) summer. The likelihood of the evidence given an average (0) summer, however, is somewhat higher, because summers with a few chilling phases could also result in descriptions of a good harvest, although these chilling phases would probably still have been described in one

source or another. The evidence is somewhat unlikely assuming an extremely hot (+3) summer, because normally such a summer would produce some descriptions of negative impacts. Therefore, such descriptions are most likely in the event of a warm summer (+1) or very warm summer (+2). These considerations result in the likelihood distribution in Table 2.

Besides the fact that more information from the sources and metadata can be expressed via this reconstruction method, the

Bayesian approach also has the great advantage that an absence of observations does not necessarily leave a gap in the climate reconstruction. In some cases, the Bayesian approach can actually turn the absence of observations into useful evidence. Most observers during the Middle Ages and early modern period were systematically more likely to notice and record extreme conditions than average ones. Thus, we may infer that the likelihood of finding no weather descriptions for a particular season in a period and region with good coverage by historical sources is higher assuming average conditions than extreme conditions.

Using values in the Pfister index, such a likelihood distribution for $\mathbf{p}(No\ observation \,|\, Climate)$ might be described by Table 3. Of course, the likelihood distribution for finding no observations given each climate value will depend on the specific sources, region, and period under study. For instance, the lack of any description of frozen lakes during some particular winter would be much higher assuming climate values of -1, 0, +1, +2 and +3 than values of -2 or -3 on the Pfister scale.





### 3.3 Prior generation

Prior seasonal temperature and precipitation states have been converted into probability distributions of Pfister indices by means of a forward model that classifies the members of a certain model ensemble into different quantiles following the Pfister index probability distribution function (Fig. 3). Pfister indices have been defined for each member as duo-decile intervals taking into account the relative severity and magnitude of the climate values (e.g., for the temperature field, each index has been associated with a range of temperatures from extremely cold to extremely warm that depends on the climatology of the

ensemble member). In the case of the HIPPO ensemble, 12-quantiles have been obtained with simulations ranging from 1420 CE to 1781 CE, while for the CESM-LME, these quantiles have been calculated with simulations from 850 CE to 1849 CE. Seasonal outputs from the ensembles have subsequently been converted into Pfister indices depending on their value, where cooler (wetter) conditions have negative indices and warmer (drier) conditions are associated with positive indices. Prior probability distributions are obtained by dividing the number of members within a given index by the total number of members in

the ensemble. For instance, if for a given season and year, 20 members of the HIPPO ensemble have an index of -2, as this ensemble has a total of 40 members, the prior probability of that index (i.e., very cold) would be of 50%. It is noteworthy to mention that the robustness of prior probability distributions depends on the number of ensemble members, recommending the use of large multi-member ensembles (such as the HIPPO ensemble) for this task.

### 3.4 Bayesian reconstruction

Once the prior and historical information are expressed in terms of probabilities, the use of Bayes' theorem to obtain the posterior Pfister distribution is straightforward (Eq. 1). The prior is multiplied by the probability distribution of the historical archives and divided by the normalizing constant (Eq, 2). This posterior probability distribution of Pfister indices represents the updated state of the climate after data assimilation. Furthermore, climate variables have also been reconstructed for each

member of the ensemble following Eq. 3 as the weighted average of mean values associated with each index, where posterior probability distributions are used as weights.

$$Climate = \sum_{n=-3}^{3} \mathbf{p}(Climate \mid n) \cdot \overline{Climate_n} \tag{3}$$

where $Climate$ is the reconstructed climate variable (i.e., $\mathrm{temp2}$, $\mathrm{TREFHT}$, or $\mathrm{wetdaysinmonth}$), $n$ is a Pfister index, $\mathbf{p}(Climate \mid n)$ is the posterior probability of $Climate$ given $n$, and $\overline{Climate_n}$ is the mean value of $Climate$ associated with

$n$. Note that as $\overline{Climate_n}$ is estimated as the mean of $Climate$ values (e.g., temperature) related to a certain Pfister index, this reconstruction method is optimal for climate variables that present a Gaussian distribution.





## 4 Results

Posterior probability distributions have been obtained for temperature-related fields from independent GCM. Figures 4 and S1 show (for summer and winter) how two different background states (top panels) are updated by information from documentary data (middle panels) to generate reconstructions with similar posterior probability distributions (bottom panels). This indicates that the information contained in historical archives can be efficiently transferred into climate reconstructions, yielding consistent results from different prior states. One of the main features of this methodology is the important influence of documentary data to shape the posterior distribution, which is based on the fact that likelihood distributions of historical texts are better defined than the prior probability distributions from models, and we therefore obtain more well-defined posterior probability distributions when we combine the two using Bayes' theorem. This is consistent with the fact that ensemble members come from different model realizations, generating a diverse range of atmospheric background states in contrast with better defined information of the real climate system extracted from documents.

On the other hand, Fig. 5 shows how precipitation-related fields such as wet days in a month can also be reconstructed using this method. Note that in this case we have complete temporal information for summer (Fig. 5b), which allows for the study of climate responses to important events such as the volcanic eruption in 1452 CE (see Subsection 5.1). Focusing on the probability distributions of Pfister indices, the same behaviour is observed: well-defined information retrieved from documentary data shapes the posterior distribution. This is even more evident than in the temperature reconstruction because precipitation is a dynamical field that depends on the internal variability of the climate system, which can differ significantly among the members of the ensemble, generating different background states and therefore providing a not-so-well-defined prior.

### 4.1 Temperature reconstruction

Seasonal 2-meter air temperature series have been reconstructed for the Burgundian Low countries using the posterior probability distributions of Pfister indices. Figure 6 shows the temperature range for each season obtained with 40 members of the HIPPO ensemble reconstructed from 1420 CE to 1499 CE. Although most of the information from documentary data is only available for summer and winter, consistent results are obtained for each season with minimum mean temperatures of $\sim 2.7$ °C in winter, maximum mean temperatures of $\sim 17.5$ °C in summer, and milder air temperatures of $\sim 7.5$ °C and $\sim 10$ °C in spring and autumn, respectively. Seasonal uncertainties defined as the difference between the maximum minus the minimum values of the ensemble for each year is quite similar for all seasons, with mean values of 0.25 °C for summer, spring and winter, and 0.4 °C for autumn.

The same air temperature reconstruction has been performed using the 13 members of the CESM-LME since 1400 CE. Figure 7 illustrates the seasonal mean temperature distribution from 1420 CE to 1499 CE for the HIPPO ensemble (solid violins) and the CESM-LME (dashed violins) after data assimilation. Similar mean temperatures are observed between both





ensembles with slightly colder winters and warmer summers when the HIPPO ensemble is employed. This indicates that although consistent results are obtained with different ensembles, the methodology preserves the internal variability of the models, transferring the background state into the final reconstruction. Furthermore, there is a significant increase of correlation between the HIPPO and CESM-LME series after assimilating the information from historical archives. Table 4 shows the seasonal Pearson correlations between the two air temperature priors from HIPPO and CESM-LME, as well as the correlations

of the posteriors. Low correlations are obtained for the priors with a maximum correlation of 0.25 in summer and a minimum one of 0.08 in winter (which are associated with the expected climate response to external forcings), indicating that the internal variability of model outputs differ significantly among each other. In contrast, significantly higher correlations are found for posterior reconstructions with a minimum Pearson correlation of 0.78 in summer and a maximum one of 0.85 in winter. This significant increase in correlation is associated with the assimilation of documentary data and highlights the importance of

retrieving information from historical archives to obtain a consistent reconstruction of the past climate.

## 4.2 Precipitation reconstruction

Moreover, wet days in a month have been obtained from 1420 CE to 1499 CE after including documentary data as shown in Fig. 8. Interestingly, in this case we have complete information for summer, allowing for the study of precipitation anomalies during the 15[th] century. On the other hand, there is almost no available information for autumn and first decades of spring time

(i.e., continuous available information for spring starts in 1471 CE). The reconstruction shows (on average) between 14 and 16 wet days per autumn month, 18 to 20 days in winter months (the annual maximum), 16 to 18 wet days in spring days, and 10 to 12 days in summer days (the annual minimum). Uncertainties are in general between 0 and 1 days, with their maximum appearing during winter time and their minimum during summer.

It is noteworthy to mention that Figure 8 presents a prominent anomaly after 1452 CE. There is an increase of wet days in a month from 10-11 summer days to 13-14 summer days that it is persistent for the 4 following years (i.e., 1453, 1454, 1455, and 1456). As a matter of fact, those years are in the top 10% of wettest days from 1420 CE to 1499 CE as depicted by the red violin in Fig. 9, increasing the number of wet days in a month to the same extent as autumn wet days. Therefore, a major event such as a volcanic eruption should have taken place during 1452 CE or 1453 CE (Robin et al., 1994; Gao et al., 2006) to

generate this strong response of the climate system (see Subsection 5.1).

## 5 Discussion

Consistent climate reconstructions have been obtained by assimilating qualitative information from documentary data compiled by historians. Although further methodological improvements can be made (e.g., developing a reconstruction method able to work with variables following non-Gaussian distributions), the Bayesian approach presented herein has been proven

effective in the task of providing a robust framework not only by retrieving climate information from historical archives, but also by taking into account uncertainties such as source reliability, data quality, as well as, regional impact. In this sense, the





methodology has been adapted to the historian work-flow (i.e., the common procedures followed in social sciences) while providing quantitative results for climate scientists, serving as a common ground where collaborative efforts between social and applied sciences can be made to improve our understanding of past climatological events.


Moreover, our method addresses larger epistemological concerns in historical research. Although scholars of history and the historical sciences have often understood their work as representing the past or debating interpretations of historical evidence and events, philosophers have raised doubts about the completeness and objectivity of such "representation" or "interpretations" (as discussed in e.g., Kuukkanen, 2015). A more defensible and productive epistemic stance could be to understand historical

research as Bayesian abductive inference - that is, a probabilistic reasoning from effects to causes, or more specifically from the traces left by the past, including physical and written records, to hypotheses about the past itself. Previous studies have demonstrated that this stance encapsulates the implicit principles underlying historical research, and that consciously adopting Bayesian abductive methods improves the accuracy of historical studies and communication (Tucker, 2004; Lavan, 2019). Our study instantiates this epistemological shift in the interdisciplinary field of historical climatology.

**5.1  Summer precipitation after volcanic eruption**

One of the most interesting aspects of the Bayesian method resides in the possibility of studying past anomalous events from documentary data recorded at the time of the incident, and reconstructing the posterior climate response at local and regional scales. In this case, we focused on summer rainfall anomalies after 1452 CE that remained persistent for the four following years as described in Subsection 4.2. To generate such climate response, the event that took place during 1452 CE must have

been of colossal proportions, which is consistent with a major volcanic eruption reported in previous studies (e.g., Robin et al., 1994; Gao et al., 2006; Raible et al., 2016; Esper et al., 2017; Hartman et al., 2019). This regional increase in wetter summers over the Burgundian Low Countries after strong volcanic eruptions could be explained as a remote effect of weakened atmospheric circulation systems such as the African and Indian monsoons (e.g., Oman et al., 2006; Colose et al., 2016; Fadnavis et al., 2021). Wegmann et al. (2014) illustrated possible mechanisms that relate weaker monsoons with positive summer pre-

cipitation anomalies over southern and central Europe after major volcanic eruptions. Within this context, weaker monsoons can lead to a weakening of the northern branch of the Hadley cell that can perturb the atmospheric circulation over the Mediterranean and North Atlantic Ocean, increasing the European summer rainfall for the incoming years after the eruption. Note that these wetter anomalies are also observed (with higher uncertainties) in priors of 1453, 1455, and 1456 (Fig. 8), indicating that GCMs are able to reproduce the forced signal after major climate events in agreement with historical records. Nevertheless,

more consistent results are obtained after data assimilation such as in the reconstruction of 1454, where no clear wet conditions were extracted from the prior, but they are clearly present in the posterior probability distribution.

It is noteworthy to mention that the consequences of this large volcanic eruption are more evident in the results presented herein than in previous reconstructions (e.g., Camenisch, 2015b). This is due to the fact that in the 1450s and 1460s, the data





quality and quantity in the Burgundian Low Countries are not very consistent with regard to weather reports in narrative sources.

Indeed, in other European regions with somewhat better source coverage in these two decades, the rainy summers resulting from the 1452 AD volcanic eruption can be traced in historical sources. For example, reports from 1454 and 1456 confirm summer precipitation and temperature anomalies in northern Switzerland: In 1454, a period of frost occurred at the end of May, during which snowfall was observed in the Black Forest to below 1150 meters above sea level. The low temperatures destroyed everything that grew in the fields. In the first third of July it was still so cold that the "living rooms" had to be heated and it rained unusually heavily. Cold and rain continued in August and September. From the summer of 1456 it is reported that from the end of June to the end of October not four days of summer weather occurred in a row and there were not 14 dry days. In addition, repeated thunder- and hailstorms devastated the area. Because it rained all the time, the crops were destroyed and the wine was sour (Camenisch, 2022). The descriptions of these two years suggest a "year without summer" in 1454 and maybe even in 1456.

This is an example of the potential of the Bayesian approach especially in the reconstruction of precipitation anomalies. With conventional methods, it would not be possible to use historical sources from Switzerland as evidence for a Pfister-index values in the Low Countries. Thus, this approach also represents an opportunity to make precise and reliable statements with few historical sources spread over a larger geographical area.

## 6   Conclusions

With the Bayesian approach applied here, we were able to integrate probabilities and uncertainty into the proven Pfister indices in an existing temperature and precipitation reconstruction of the Burgundian Low Countries during the 15[th] century. In this process, several gaps could be closed, because even years with ambiguous, little or no concrete information from historical sources could be evaluated. This was possible because no explicit decisions had to be made for an index value, but all index values were evaluated according to their probability of occurrence. Such an assessment is possible thanks to a precise knowledge of the historical sources and the way weather-related information is presented in them. The Bayesian reconstructions obtained herein are not only physically consistent due to background states of the climate system provided by last generation model simulations, but also historically accurate thanks to the information retrieved from historical documents. Hence, this study opens the door for the assimilation of documentary information contained in the archives of most History faculties around the world, motivating the collaboration between social and applied sciences.

*Data availability.* https://doi.org/10.6084/m9.figshare.17088911.v1





*Author contributions.* SW developed the idea of applying this methodological approach to historical climatology. SW, CC, SB, FJS, and

335   QP have worked on further developing and adapting this approach. CC used her sources to create a new reconstruction for the Netherlands, which expresses probabilities. FJS, RH, and SB did the extensive calculations on the models.

*Competing interests.* One author is member of the editorial board of Climate of the Past.

*Acknowledgements.* The authors would like to thank PAGES (Past Global Changes) for supporting the CRIAS working group in this research. This work was supported by the cogito Foundation (Project Nr. 20-108-R) and by the European Commission (ERC Grant PALAEO-RA,

340   787574).





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





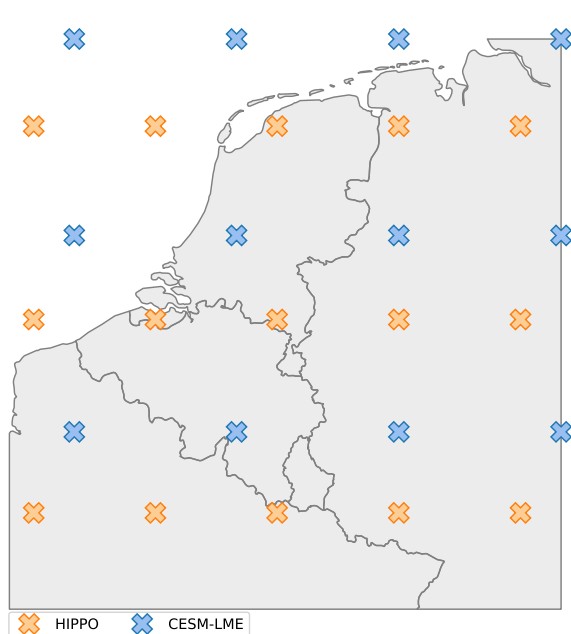

**Figure 1.** Map of the Burgundian Low Countries with historical documents describing past climate events during the 15$^{th}$ century. The region is defined as the land area delimited by latitudes between 1.5°N and 10°N, and longitudes between 48.5°E and 54°E. Orange and blue crosses depict the grid points where temperature and precipitation series have been extracted from the HIPPO ensemble and the CESM-LME, respectively.

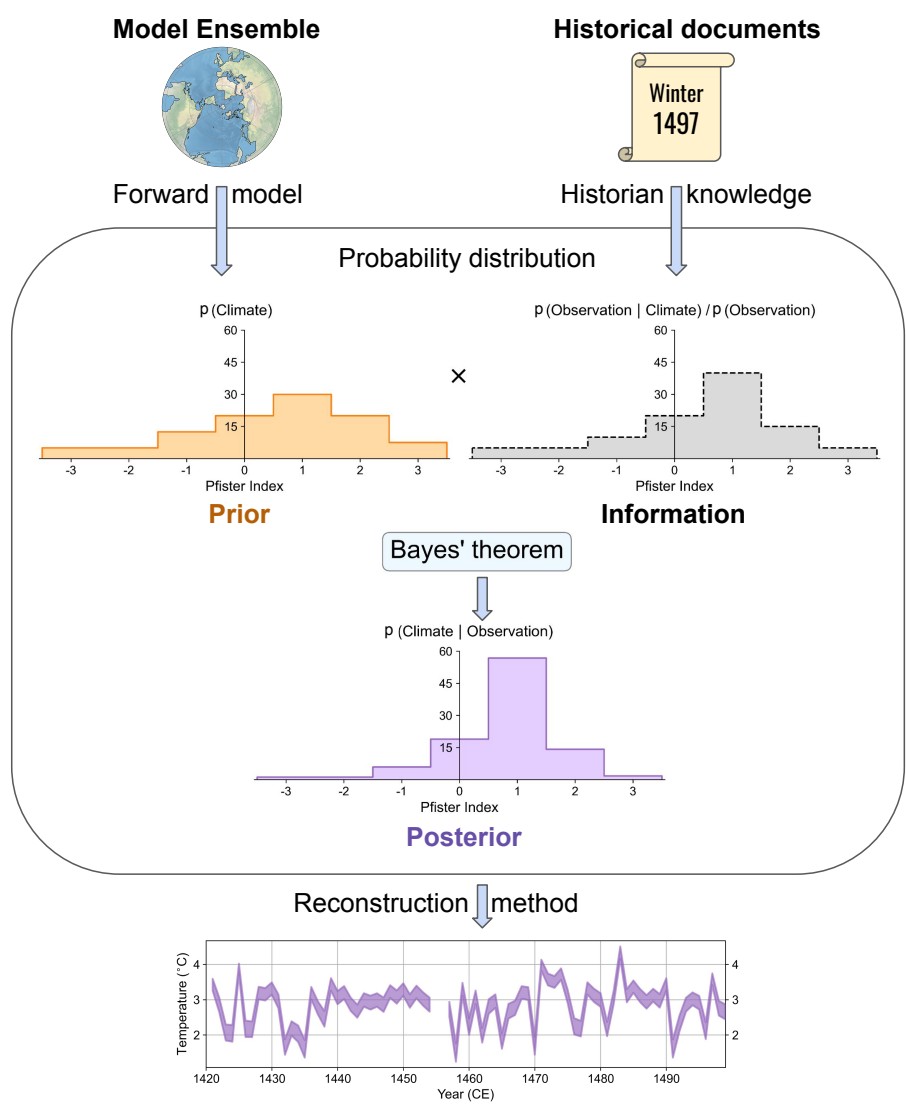

**Figure 2.** Methodology followed to assimilate historical documents into seasonal temperature and precipitation reconstructions of the Burgundian Low Countries during the 15th century. Outside the rounded square are the inputs (top) and output (bottom) of the climate reconstruction. All inputs are converted into probability distributions of Pfister indices from which posterior probability distributions are obtained using Bayes' theorem.





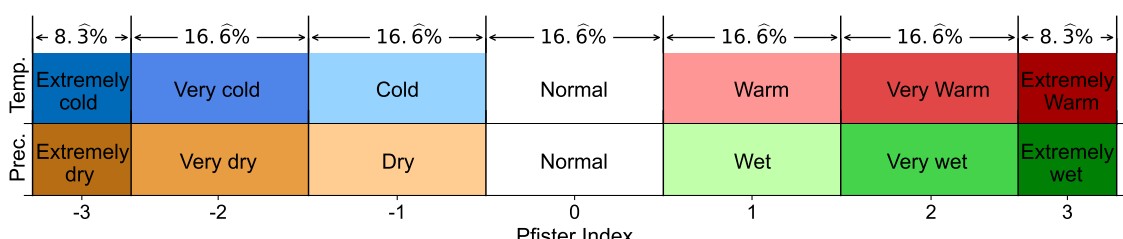

**Figure 3.** Pfister indices for temperature (top) and precipitation (bottom). Percentages show how climate information is distributed in terms of its severity. Note that Pfister indices classifies temperature and precipitation values into 12-quantiles (or duo-deciles) with an uniform distribution of 2 duo-deciles for each index between -2 and 2, while minimum and maximum indices (i.e., -3 and 3) are associated with the lowest and highest quantiles, respectively.



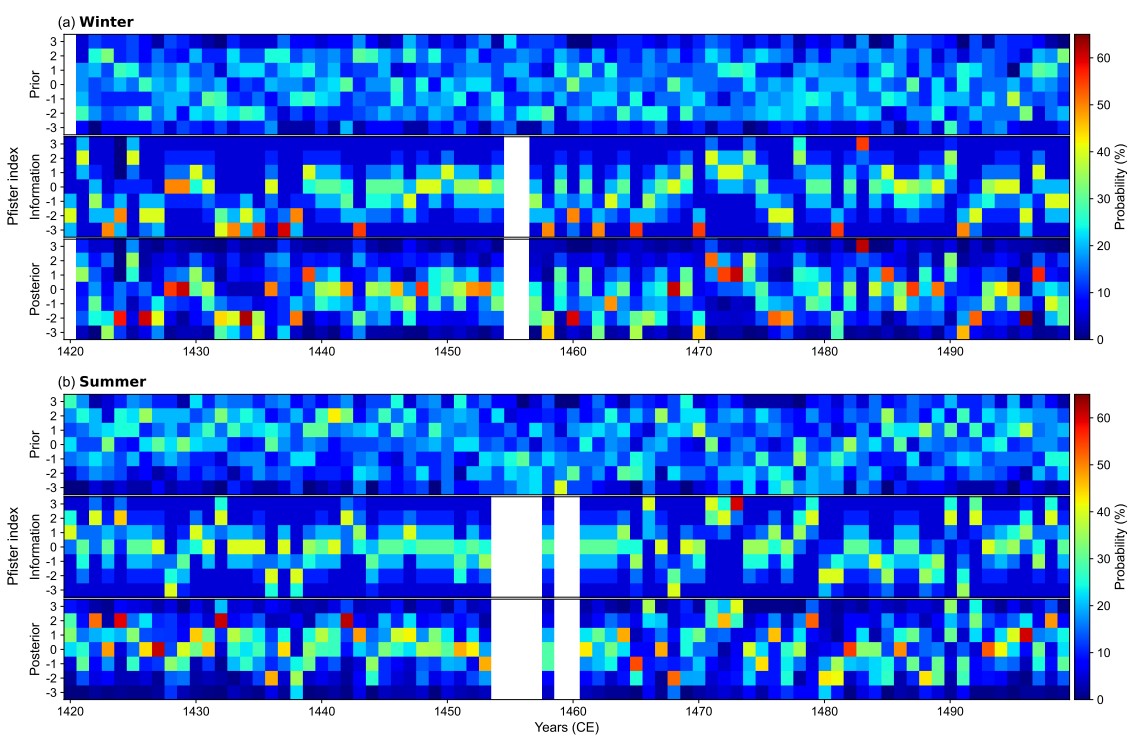

**Figure 4.** Prior, historical information, and posterior probability distributions of seasonal temperatures in the Burgundian Low Countries for (a) winters and (b) summers ranging from 1420 CE to 1499 CE. Top panels represent the prior probability distribution, p(Climate), extracted from 2-meter air temperatures of the 40-member HIPPO ensemble and converted into Pfister indices. Middle panels illustrate the probability distribution of the information, p(Observations | Climate)/p(Observations), acquired from historical archives compiled and converted into Pfister indices by historians. Bottom panels depict the posterior probability distribution, p(Climate | Observation). White spaces are shown when there are no observations available and therefore the posterior matches the prior.



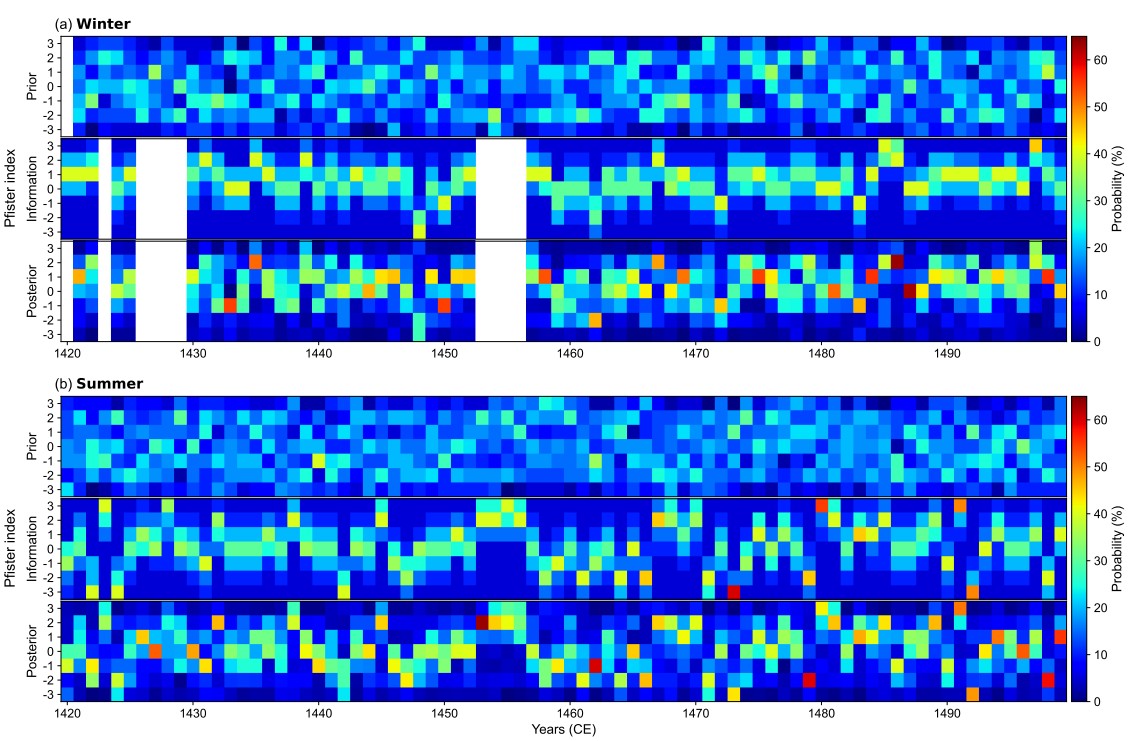

**Figure 5.** Prior, historical information, and posterior probability distributions of seasonal wet days in a month in the Burgundian Low Countries for (a) winters and (b) summers ranging from 1420 CE to 1499 CE. Top panels represent the prior probability distribution, p(Climate), extracted from the 40-member HIPPO ensemble and converted into Pfister indices. Middle panels illustrate the probability distribution of the precipitation information, p(Observations | Climate)/p(Observations), acquired from historical archives compiled and converted into Pfister indices by historians. Bottom panels depict the posterior probability distribution, p(Climate | Observation). White spaces are shown when there are no observations available and therefore the posterior matches the prior.



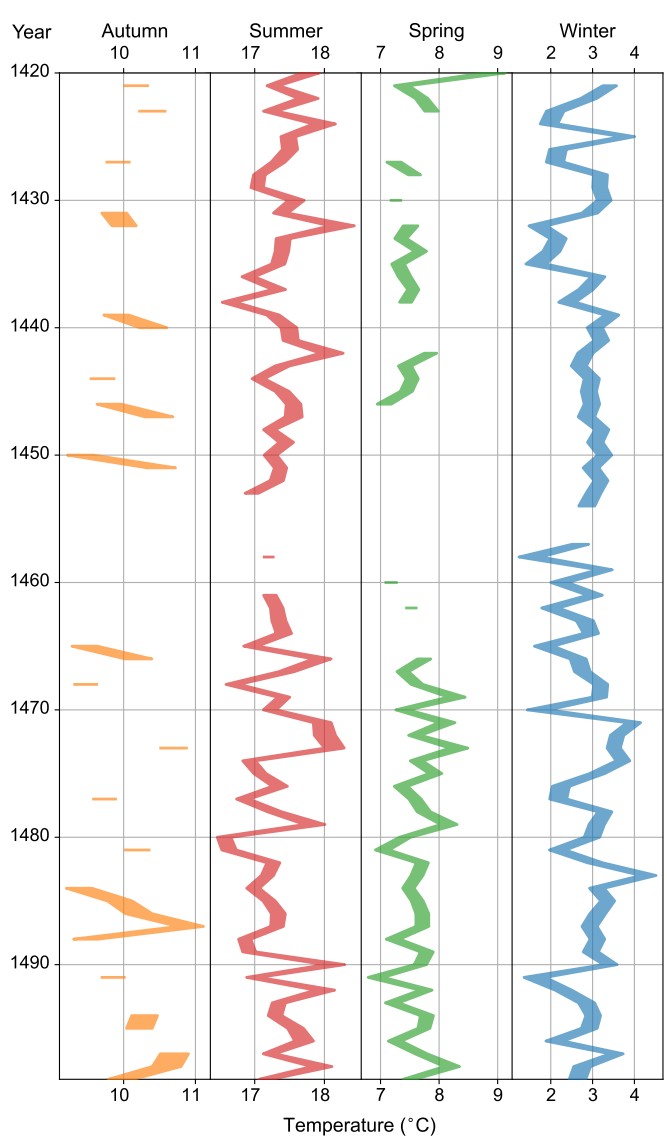

**Figure 6.** Seasonal mean temperatures in the Burgundian Low Countries from 1420 CE to 1499 CE. Coloured areas show the range of temperatures of the HIPPO ensemble reconstructed from posterior probability distributions after assimilating regional information from historical documents.





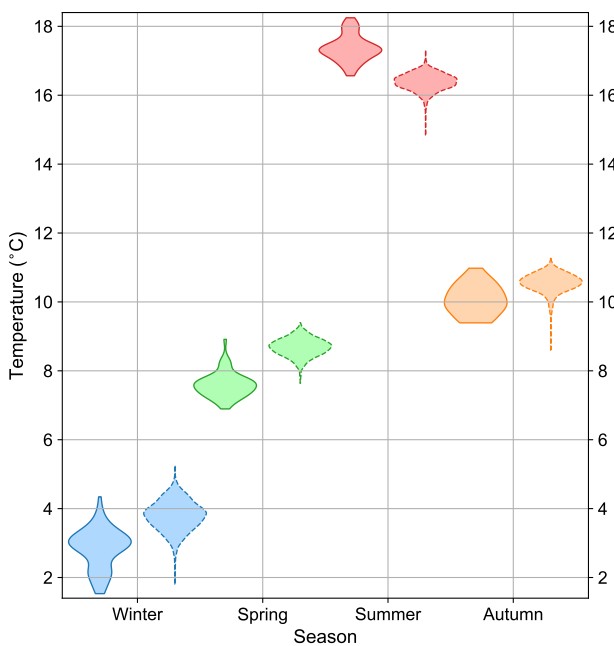

**Figure 7.** Distribution of mean temperatures in the Burgundian Low Countries from 1420 CE to 1499 CE after data assimilation. Solid (dashed) violins show the seasonal distribution when 2-meter air temperatures from the 40-member (13-member) HIPPO ensemble (CESM-LME) are employed to generate the prior.



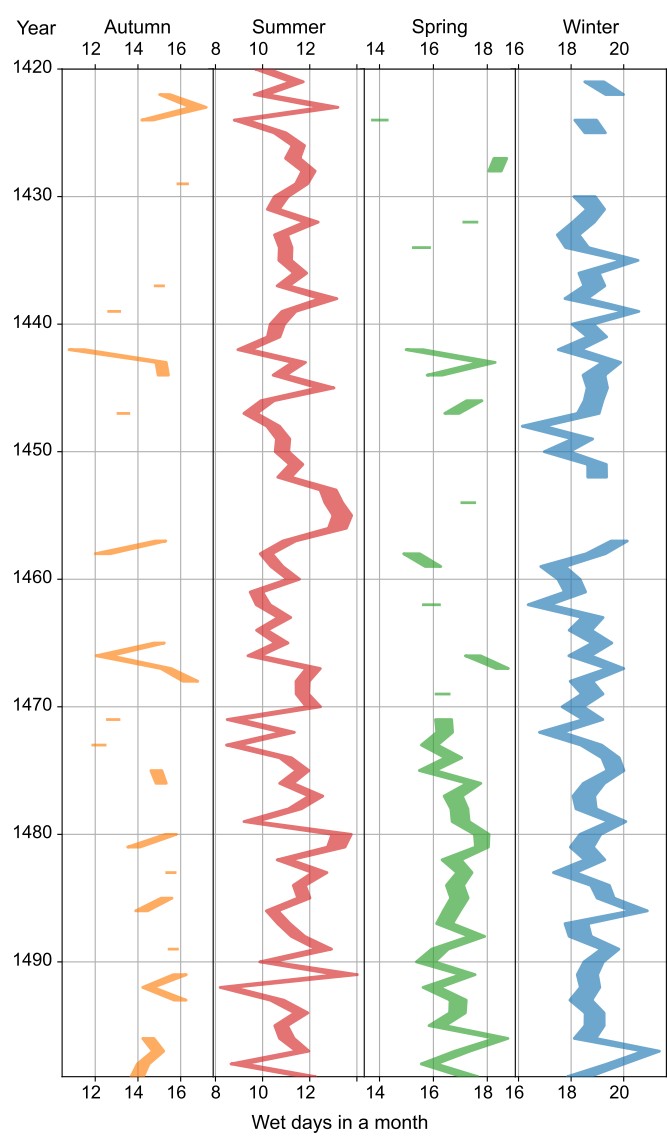

**Figure 8.** Seasonal mean wet days in a month in the Burgundian Low Countries from 1420 CE to 1499 CE. Coloured areas show the range of wet days in a month of the HIPPO ensemble reconstructed from posterior probability distributions after assimilating regional information from historical documents.

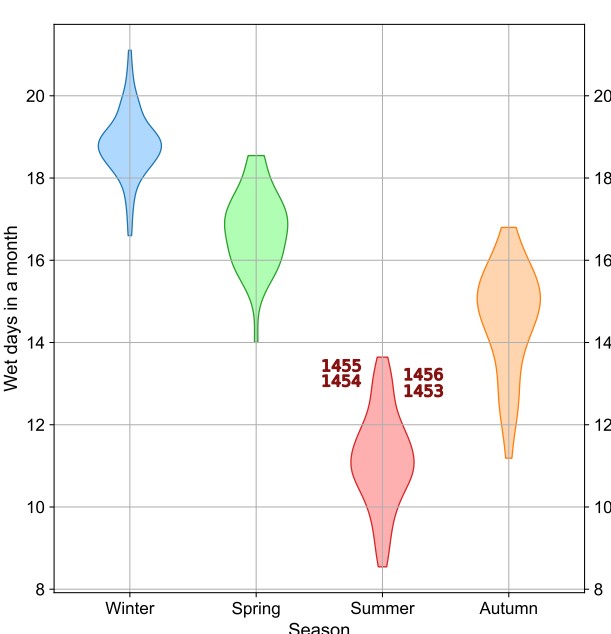

**Figure 9.** Distribution of average wet days in a month over the Burgundian Low Countries from 1420 CE to 1499 CE after data assimilation using the HIPPO ensemble as the prior. Depicted in red are the 4 summers following a strong volcanic eruption in late 1452 CE (or early 1453 CE).



**Table 1.** List with information concerning the model ensembles employed to generate the background state of the atmosphere. From left to right are the ensemble and model names, the number of members, the pre-industrial time period with available information, and the spatial resolution in decimal degrees.

| Ensemble | Model | Members | Period (CE) | Resolution (°) |
|----------|-------|---------|-------------|----------------|
| HIPPO | ECHAM 6.3.05p2 | 40 | 1420 - 1781 | 1.9×1.9 |
| CESM-LME | CESM 1.1.2 (CAM5) | 13 | 850 - 1849 | 1.9×2.5 |

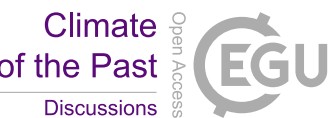

**Table 2.** Example of the distribution of likelihoods of a well-documented very warm summer.

| Pfister index | -3 | -2 | -1 | 0 | 1 | 2 | 3 |
|---|---|---|---|---|---|---|---|
| **Likelihood** | 0.05 | 0.05 | 0.05 | 0.15 | 0.25 | 0.30 | 0.15 |





**Table 3.** Example of the distribution of likelihoods of an undocumented summer during a phase with otherwise good source density.

| Pfister index | -3 | -2 | -1 | 0 | 1 | 2 | 3 |
|---|---|---|---|---|---|---|---|
| **Likelihood** | 0.05 | 0.05 | 0.20 | 0.40 | 0.20 | 0.05 | 0.05 |



**Table 4.** Pearson correlation between the HIPPO ensemble and the CESM-LME multi-member mean temperatures over the Burgundian Low Countries for the 1420-1499 CE period. 2-meter air temperature correlations have been calculated for each season before (Prior) and after (Posterior) assimilation of historical documents over the region.

| Season | Prior | Posterior |
|--------|-------|-----------|
| Winter | 0.08 | 0.85 |
| Spring | 0.18 | 0.81 |
| Summer | 0.25 | 0.78 |
| Autumn | 0.21 | 0.84 |