# Peer review of "A Bayesian Approach to Historical Climatology for the Burgundian Low Countries in the 15th Century"

_Climate of the Past, 2021_

## Author Response (AR1)

**Reply to anonymous referee #1**

General comments

The manuscript is an interesting work on climate reconstructions based on historical documents. It is an original approach from a methodological point of view. The Bayesian approach is an interesting methodology, and an original way to integrate the output of climate models (prior state) with historical data (observation/climate conditional probabilities). In this sense, it seems very appropriate to be published In "Climates of the Past". However, I have some doubts, questions, and suggestions.

Many thanks for your referee report, which we appreciate very much.

Specific comments

1. Although the authors include some references on documentary data, it is desirable a more complete and detailed description of data sources (authors, motivations, spatio-temporal coverage, density of information, etc.). Please, include in Figure 1 the cities and/or locations with historical information.

Many thanks for this comment. The original data set is already described and published in a paper in Climate of the Past and in a monograph (Camenisch 2015a, 2015b) to which we give reference. We added more information on the sources to the paper and the major places to a map.

2. According to the authors, the reconstruction method is strictly applicable using variables following Gaussian distributions. In fact, Pfister indices use implicitly this hypothesis assuming the symmetry around the 0 value. This may be true in the case of temperatures, but I have doubts on precipitations, where it may be not appropriate to define symmetrical indices (from -3 to +3), due to the non-gaussian character of rainfalls.

This is correct. Our method is only applicable to variables that follow a Gaussian distribution such as temperature. This would advise against using our method to reconstruct precipitation due to the non-gaussian nature of rainfalls. To tackle this issue, we did not reconstruct the usual precipitation rate, but a Gaussian-distributed custom variable (wetdaysinmonth) defined as the number of days in a month with more than 1 mm precipitation within a model day (lines 104-106 of the manuscript). This prevented us from reconstructing any non-gaussian variable, keeping the study free from inconsistencies. We will clarify this point in the revised version of the manuscript.

3. Tables 2 and 3. The assignment of likelihoods to Pfister indices is arbitrary. In the case of Table 2, why 25 to index +1, and 0.30 to index +2, and not the opposite, 0.30 to index +1 and 0.25 to index +2? Indices methodology tries to convert qualitative descriptions into numerical values, and, certainly, some degree of subjectivity is always present. I recognize the effort of this approach to reduce this ambiguity, but this example do not diminish my doubts about this problem.

Many thanks for this comment. Table 2 shows an example of a well-documented summer, whose descriptions are similarly – but not identically -- likely assuming either a +2 (very warm) and +1 (warm). Although +1 seasons are a priori more probable than +2 seasons, that is not our concern in this step of the reconstruction. In this step of the reconstruction, we are assessing which hypothetical climate state would have been more likely to produce the present evidence. In this case, it is slightly less likely that a hypothetical +1 summer rather than a +2 summer would have produced these descriptions, since summers that were fairly normal (only +1) tended to result in less description of warm conditions. Nevertheless, if the a priori probability of a +1 summer is much higher than a +2 summer, then the posterior probability distribution will reflect this.

In the original index, such a case would appear simply as +2 with no possibility to express these uncertainties about the evidence or prior probabilities. Thus, even though both the index method and the Bayesian approach rely on expert judgement, we believe the latter is less arbitrary.

This is just one example. The entire reconstruction is based on the 450-page PhD thesis Camenisch 2015a, where each year is examined individually. We changed the caption to "well documented summer with a similar distribution of likelihoods for +2 (very warm) and +1 (warm)" to make it clearer. We added to the text that "well-documented" means there are several descriptions of the weather in the historical sources.

The model example in Table 3 is different: In this case, there are no descriptive sources for this summer. Since descriptions are available for very many summers of the 15th century and since many decades have a very good coverage by historical sources as a whole (in the same year, the summer may not be described, but maybe the spring and the autumn are), it is practically impossible that extreme events would not have left any traces. So, there is maybe no description of the weather in the sources, but of course this is not equivalent to no information about the season. We changed the captions to "no description of the weather" instead of "undocumented".

4. In relation to Table 3 and the lack of information, the absence of information it is not equivalent to the absence of climate events (extremes) in the past. It depends on the nature of data sources, spatio-temporal coverage and resolution (it is possible to find new data sources that compel to refine the reconstructions). Therefore, it is important not only the description of data sources (Point 1), but also the study of their spatio-temporal coverage, that is, their density of information (distribution of reports according spatial and temporal scales).

This is exactly what we want to say with this hypothetical example. Our knowledge does not only consist of the lack of a description of this one summer. We know much more, of course. For example, we have detailed knowledge of these sources in the Burgundian Low Countries and can therefore make a very good estimate of the data coverage for each decade or even for each individual year: e.g., whether for the years in question there were one or more reliable chroniclers reporting regularly; whether there where important political events that could have suppressed weather reports; and where there are indicators such as grain prices show for the region and season. I suspect that the referee was bothered by the term "undocumented" in the caption. We adjusted this caption go more into detail in the text about what exactly we mean by this example.

5. How do you calibrate and/or validate your reconstruction? This is the major problem that I see in this manuscript. Criteria on uncertainty and/or error bars are unclear for me. A more detailed description on technical aspects of this methodology would be desirable.

Thank you very much for this important comment. In this present reconstruction, the highest probability is in each case on the same index values that were already determined in 2015. So these maximum values are not new. New is the addition of (lower) probabilities each additional index value that was not selected in 2015. The detailed explanations why originally one index value was selected and not another, as well as the criteria for each index value, can be found in Camenisch 2015a. In Camenisch 2015b, the indices were compared with other reconstructions (e.g., Litzenburger 2015) and examined for correlations. These methodological approach was 2015 accepted in the peer review process of this journal. In the revised version we explain this more clearly.

For this region and period (15$^{th}$ century), there are no historical climatology index series that overlap with instrumental records to enable a calibration-verification procedure. Camenisch 2015a and 2015b created an index series that was accepted based on established methodology,

comparison with series for neighboring regions, and comparison to reconstructions based on paleoclimate proxies.

Our method builds on the research in Camenisch 2015a and 2015b. It incorporates all the evidence and knowledge utilized in that reconstruction. However, it removes implicit judgments concerning prior probabilities that are built into the traditional index method and makes explicit those judgments regarding likelihoods that are also built into the index method. Therefore, it should be more reliable and less arbitrary than the traditional index method.

Moreover, our method incorporates all available information into a single posterior probability distribution for the target variables. Previously, scholars may have examined reconstructions from paleoclimate reconstructions or climate field reconstructions that integrated climate forcings and paleoclimate proxies alongside historical climatology reconstructions. They may then have made their own informal inferences regarding the most probable true values – a kind of fuzzy Bayesianism. We formally employs Bayes' theorem, integrating all information to obtain single series of posterior probability distributions.

Because our method already integrates all available information, and because there is no overlapping instrumental record, the resulting posterior probability distributions cannot be compared to any other independent measure or reconstruction. Therefore, we cannot demonstrate its reconstruction skill. Instead, this study demonstrates: (1) that the method is feasible and can fully integrate information from climate modeling and assessment of historical records; (2) that historical knowledge can be modeled as a process of Bayesian abductive inference; and (3) that the weather and climate information in historical records -- expressed as a ratio of likelihoods $p(e|h)/p(e)$ -- can bring convergence to divergent paleoclimate model outputs.

We revised the introduction to clarify the scope of the article.

6. Finally, I miss an adequate comparison with other reconstructions. In particular, to obtain a clear view of the convenience of this approach, it would be interesting a comparison with the simple reconstruction based on Pfister indices. In addition, it would desirable to find reconstructions from other proxy data (in particular tree rings), to validate your reconstruction, or, at least, to compare your results with those from other proxy data.

Many thanks, see answer 5 above. In regard to the tree rings: for methodological reasons, there is no separate tree ring reconstruction for this area.

**Reply to anonymous referee #2**

In their manuscript, Camenisch et al. present a new approach to quantitative reconstruction of temperature and precipitation characteristics from documentary sources. Through Bayesian inference, categorical data derived from historical archives are assimilated into GCM-generated ensembles of climate simulations, effectively combining temporal variability of both these sources. Application of the technique is demonstrated for seasonal temperatures and numbers of days with precipitation in the Low Countries (NW Europe), over the 1420-1499 CE period.

The paper is competently written and topically well suited for the 'International methods and comparisons in climate reconstruction and impacts from archives of societies' special issue of the 'Climate of the Past' journal. I only have a few comments/suggestions regarding the methodology, results, and their presentation (I leave it at authors' discretion whether and how they will consider them in preparation of the final manuscript):

Many thanks for your referee report and your comments!

(C1) Extensive ensembles of GCM simulations were used to generate the base (prior) probability distributions. However, since the year-to-year variability in such simulations is largely uncorrelated with historical variability in the climate system, retaining full intra-ensemble variability (as described in Sect 3.3) seems to add unnecessary noise to the prior data. This noise is then partly carried over to the posterior data (this is especially apparent in Fig. S1, visualizing results obtained for the smaller (13-member) CESM-LME ensemble). Perhaps using somewhat less 'noisy' data to generate the prior probability distribution (e.g. by employing mean value of the ensemble instead of its complete spread) would result in less noisy reconstructions, while still retaining the relevant variability from the GCM-simulated series (such as components tied to boundary conditions and external forcings, which are shared by all ensemble members).

Many thanks for this comment. We understand that GCM internal variability has usually a low correlation with historical variability. However, climate simulations are physically consistent, and therefore they are well suited to describe the background state of the atmosphere prior to any observation. For instance, GCM outputs are especially good at capturing the climate response to external forcings such as volcanic eruptions, which can further improve the reconstruction of post-eruption years. In this sense, a large ensemble of GCM simulations is needed to calculate the prior probability distribution. Note that if only the mean value of the ensemble is used, the prior probability distribution of Pfister indices would only have one value with a likelihood of 1. Moreover, the historical variability is embedded into the posterior probability distribution through the information provided by historians, yielding high correlations among reconstructions with priors generated from independent GCM ensembles (HIPPO vs CESM-LME) with completely different internal variabilities (Table 4).

(C2) Minimum probability threshold of 0.05 was prescribed when generating the probability distributions (l. 154+). It feels that in some situations, this may act as unnecessary artificial degradation of the signal (e.g., when a distinctly hot summer is indicated by the documentary sources, yet the probabilities for sub-normal temperatures are still set to be greater than zero regardless). Perhaps using a simple formal parametric approximation of the probability function, e.g. by (suitably transformed) binomial or Gaussian distribution, would better capture the related uncertainties (with probability values outside of the most likely categories still being non-zero, but not constrained by an arbitrary constant).

The initial idea was not to include any threshold to the observational likelihoods, as suggested by the reviewer. However, this led to a posterior probability distribution highly governed by the observations, removing most of the prior information provided by climate models. Note that, for observational likelihoods close to 0, the posterior probability is also close to 0, regardless of the prior probability. Therefore, a consensus between historians and climate scientists was achieved to set a minimum threshold of 0.05. The 0.05 is based on the experience with the relevant historical sources, and not only statistical convenience. Sources of the period may contain copying errors and incorrect dates. Thus, even for a cold summer, we might have a description of great heat: not because observers were incapable of telling hot from cold, but because we have a description for the wrong year or location. And it allows for the partial propagation of information from model outputs into the posterior probability distribution. In future studies, we may consider whether the correct likelihood should be more like 0.2 or 0.3, but we chose to err of the side of caution and simplicity in this first study. We will clarify this point in the revised version of the manuscript.

(C3) I wonder about uncertainties/ranges shown for the reconstructions in Figs. 6 & 8 and how they relate to the posterior data visualized in Figs. 4 & 5. For instance, in Fig. 4a (winter temperature), the 1459 CE temperature estimate seems quite uncertain (i.e., the posterior probability distribution is rather widely spread among several categories), whereas much lower uncertainty is indicated for 1460 CE (narrower probability distribution, dominated by a single category). Yet, there is no major difference in the size of the estimated temperature ranges for these years in Fig. 6 (in fact, the ranges seem to be near-identical in size throughout the entire period covered). If these are derived solely from min-max values of the GCM ensemble (as described at l. 233+), perhaps it would be useful to also provide uncertainties derived from the spread of posterior distributions in their entirety (and thus to consider not only uncertainty of the prior (GCM-based) data, but also that from the documentary sources).

The reviewer is right and uncertainties derived from the spread of posterior distributions will be included in Figs. 6 & 8. The same way that seasonal mean temperatures were calculated using the posterior probability distribution of Pfister indices, their corresponding weighted standard deviations will also be calculated, so that reconstruction uncertainties can be better assessed.

(C4) It might be useful to see how well the temperature/precipitation reconstructions match actual weather variability typical for the target region (to see if, e.g., variance of the reconstructions matches the real climate, or if there is under/overestimation). This could be done, for instance, by adding observational distributions for the instrumental period to Figs. 7 and 9 (and discussing which eventual differences stem from comparing two different periods, and which may be related to biases in the reconstruction itself).

This is a good suggestion that can be implemented in the revised version of the manuscript. Although a direct comparison of the 15th century against the 19th-21st centuries cannot be made, an observational dataset such as HadCRUT 5 / CRUTEM 5 (land only) can be used to obtain an observational distribution of the temperature since 1850 CE, allowing for the analysis of temperature changes between those two time periods. On the other hand, a more challenging task is to find an observational data set of wet days in a month over the area of interest, and therefore we will have to rely on reanalysis for the precipitation assessment, which can be biased due to the model component of these hybrid products.

Minor/technical comments

Abstract, l. 9+: '… our reconstructions present a high seasonal temperature correlation of ~8 independently of the climate model employed to estimate the background state of the atmosphere.' – it is not quite clear from this formulation what the correlation value refers to (i.e., which two signals are being compared)

l. 188: '… (drier) conditions are associated with positive indices.' – this seems to clash with definition at l. 131, which associates positive values of the index with wetter conditions

Corrected.

l. 198: comma instead of dot

Corrected.

Fig. 9: Maybe it would be useful to add a symbol to each of the four post-volcanic years (instead of just number), so that it is more clear that these are specific data points

We hope that the years already show that it is a specific point.

Sect. 2: Perhaps elaborate a bit more on the exact extent of the target region – it might be particularly helpful to show locations pertaining to individual documentary records used, e.g. by including them in Fig. 1

We added a second map with the places where the sources come from.